# NK Cells Armed with Chimeric Antigen Receptors (CAR): Roadblocks to Successful Development

**DOI:** 10.3390/cells10123390

**Published:** 2021-12-01

**Authors:** Ali Bashiri Dezfouli, Mina Yazdi, Alan Graham Pockley, Mohammad Khosravi, Sebastian Kobold, Ernst Wagner, Gabriele Multhoff

**Affiliations:** 1Central Institute for Translational Cancer Research Technische Universität München (TranslaTUM), Department of Radiation Oncology, Klinikum Rechts der Isar, Einstein Str. 25, 81675 Munich, Germany; gabriele.multhoff@tum.de; 2Pharmaceutical Biotechnology, Department of Pharmacy, Ludwig-Maximilians-Universität (LMU), 81377 Munich, Germany; mina.yazdi@cup.uni-muenchen.de (M.Y.); ernst.wagner@cup.uni-muenchen.de (E.W.); 3John van Geest Cancer Research Centre, School of Science and Technology, Nottingham Trent University, Nottingham NG11 8NS, UK; graham.pockley@ntu.ac.uk; 4Department of Pathobiology, Faculty of Veterinary Medicine, Shahid Chamran University of Ahvaz, Ahvaz 61357-831351, Iran; m.khosravi@scu.ac.ir; 5Center of Integrated Protein Science Munich (CIPS-M) and Division of Clinical Pharmacology, Department of Medicine IV, University Hospital, Ludwig-Maximilians-Universität München, Member of the German Center for Lung Research (DZL), 80337 Munich, Germany; sebastian.kobold@med.uni-muenchen.de; 6German Center for Translational Cancer Research (DKTK), Partner Site Munich, 80337 Munich, Germany

**Keywords:** immunotherapy, natural killer cells, chimeric antigen receptor, tumor antigen, gene delivery

## Abstract

In recent years, cell-based immunotherapies have demonstrated promising results in the treatment of cancer. Chimeric antigen receptors (CARs) arm effector cells with a weapon for targeting tumor antigens, licensing engineered cells to recognize and kill cancer cells. The quality of the CAR-antigen interaction strongly depends on the selected tumor antigen and its expression density on cancer cells. CD19 CAR-engineered T cells approved by the Food and Drug Administration have been most frequently applied in the treatment of hematological malignancies. Clinical challenges in their application primarily include cytokine release syndrome, neurological symptoms, severe inflammatory responses, and/or other off-target effects most likely mediated by cytotoxic T cells. As a consequence, there remains a significant medical need for more potent technology platforms leveraging cell-based approaches with enhanced safety profiles. A promising population that has been advanced is the natural killer (NK) cell, which can also be engineered with CARs. NK cells which belong to the innate arm of the immune system recognize and kill virally infected cells as well as (stressed) cancer cells in a major histocompatibility complex I independent manner. NK cells play an important role in the host’s immune defense against cancer due to their specialized lytic mechanisms which include death receptor (i.e., Fas)/death receptor ligand (i.e., Fas ligand) and granzyme B/perforin-mediated apoptosis, and antibody-dependent cellular cytotoxicity, as well as their immunoregulatory potential via cytokine/chemokine release. To develop and implement a highly effective CAR NK cell-based therapy with low side effects, the following three principles which are specifically addressed in this review have to be considered: unique target selection, well-designed CAR, and optimized gene delivery.

## 1. Introduction

Cancer is a major health burden and mortality rates continue to increase worldwide. Despite aggressive treatment regimens consisting of surgery, radio-/chemotherapy, and small molecule/targeted therapies in different combinations, overall survival of patients with late-stage tumors remains mostly poor. Therefore, there is an urgent need for more specific and effective therapies that cause fewer complications [1]. Our immune system has a natural capacity to prevent tumor progression which involves cytokine/chemokine release, as well as antibody or cell-based mechanisms leading to cancer cell death. However, the tumor and its microenvironment have developed escape mechanisms which limit the capacity of the immune system to effectively fight malignant cells [2]. The inception of cancer immunotherapy has heralded a paradigm shift towards unleashing or reprogramming immune responses to boost the efficacy of host anti-tumor reactions. Successful examples include combatting checkpoint inhibition of T cells using blocking antibodies, and the use of bispecific ‘engager’ antibody constructs [3,4].

Adoptive cell therapy (ACT) is based on the infusion of immunologically active and tumor-specific effector cells that seek and recognize cancer cells in a patient with a therapeutic intention. ACT has evolved from bench-to-bedside due to an increased understanding of tumor biology and general immunological principles [3,5,6]. The introduction of chimeric antigen receptor (CAR) technology has enabled the adoptive transfer of immune cells to become a more practical approach [7,8]. To date, T cells have been the most commonly engineered cell type, especially by CAR [7] and the current developments in CAR T cell-based therapies have greatly improved the scope of modern, targeted cancer therapy [9]. Among others, the US Food and Drug Administration (FDA) has approved several CD19-directed CAR T cell therapeutic products for the treatment of hematological malignancies, such as types of B cell lymphomas and acute lymphoblastic leukemia (ALL) [7]. In 2021, B cell maturation antigen (BCMA)-directed CAR T cells were approved for treating multiple myeloma (MM) [10]. However, challenges originating from CAR T cell therapy such as their relatively high cost and time-consuming production, insufficient trafficking to solid tumors, induced cytotoxic effects including immune effector cell-associated neurologic syndrome (ICANS) and cytokine release syndrome (CRS), have emerged as clinically relevant challenges that can only be managed in experienced centers [11,12]. Accordingly, it is important to mitigate against these problems while safeguarding and enhancing CAR activity. Among other immune cell platforms (e.g., γ/δ T cells, NKT cells, and macrophages), natural killer (NK) cells have been considered as a potential alternative for genetic engineering with CARs [13]. CARs have been successfully engineered into NK cells, and their efficacy has been tested in preclinical and early clinical studies [8]. CAR NK cells exhibit several advantages over CAR T cells which have the potential to enhance effectiveness and safety. The first clinical use of CD19 CAR NK cells in patients suffering from relapsed/refractory lymphoid malignancies demonstrated a persistence of CAR NK cells with encouraging remission rates and clinical responses [14,15]. The high potential of NK cell-mediated killing can be related to CAR-dependent mechanisms and their ability to engage cancer cells via CAR-independent mechanisms. However, depending on the study design, the CAR NK cell product alone could not be directly compared to conventional CAR T cells, and all but one patient who responded with a complete remission had either concomitantly or subsequently received additional therapies [14]. Considering that safety is an important parameter for clinical application, it was suggested that CAR NK cells can reduce the risk for some life-threatening complications, such as severe inflammatory reactions which frequently occur upon CAR T cell infusions [15]. Importantly, due to the reduced risk of graft versus host disease (GVHD), CARs can be introduced into allogeneic NK cells and thereby provide multiple potent “off-the-shelf” sources for a safe cell-based adoptive immunotherapy [8].

Despite encouraging outcomes of CAR NK cell therapies in hematological diseases, concerns still exist regarding the low transfection capacities of NK cells, the choice of “off-the-shelf” NK cell sources, and the specificity of the CAR for target recognition, especially in antigen-heterogeneous malignancies. Several other challenges such as tumor resistance driven by antigen escape mechanisms and low infiltration rates into solid tumors can also impair the CAR therapy outcome [16,17,18]. Optimal CAR efficiency can be supported via a precise structural design paired with a rational selection of CAR targets based on the expression density of the antigen on the tumor cell. The CAR structure and surface localization on effector cells affect the binding affinity, immune synapse formation, and subsequent immune cell activation [19]. The efficiency and safety of CAR cell therapy also depend on the tumor-specificity of the chosen antigen, as the occurrence of off-target effects is mostly attributed to on-target off-tumor immune responses [20]. Moreover, an optimized CAR structure requires a carrier that sufficiently transmits the expression message. Since hematopoietic cells, such as NK cells are among the cell types which are not easy to be transfected in vitro and in vivo, systems are required which allow not only an optimized, but also safe gene transfer while keeping the costs and the burden of manufacturing under GMP conditions at a minimum [21]. 

In this review, we briefly discuss the unique properties of NK cells which support the promise of CAR NK cells in cellular immunotherapy. The review focusses on three major topics which are important for a successful CAR NK cell therapy: (i) production of well-designed CARs, (ii) selection of the correct tumor-specific/tumor-associated antigen as a target, and (iii) optimal engineering of NK cells by efficient CAR delivery.

## 2. NK Cells—A Promising Cellular Platform for CAR Engineering

The success of CAR cell-based immunotherapies is highly dependent on the biological features of the effector cell population used for genetic engineering because the CAR aims to specifically direct and boost their cytotoxicity potential. Among different types of immune cells, NK cells have gained major interest in adoptive immunotherapies given their inherent non-major histocompatibility complex (MHC)-restricted cytotoxic potential against different malignancies (Figure 1) [16,17].

NK cells, formerly termed “large granular lymphocytes”, play an indispensable role in bridging and orchestrating innate and adaptive immune responses [22], producing cytokines, and stimulating dendritic cell (DC) and B cell maturation. Phenotypically, NK cells are characterized by lacking the expression of CD3 and the T cell receptor complex, and several germline-encoded receptors. Based on the expression of the neuronal adhesion molecule CD56 and the low-affinity Fc gamma receptor CD16, NK cells can be classified into two major subsets: CD56^dim^CD16^high^ cells with high cytotoxicity and CD56^bright^CD16^low^ cells with secretory activity dominantly in the peripheral blood, and tissues and secondary lymphoid organs, respectively [23]. An efficient innate defense requires the functional harmonization of these two major subsets for the initiation and progression of immunity under pathogenic conditions [24].

NK cell responses against healthy and transformed cells are regulated by a fine balance in the expression of inhibitory and activating receptors. The surface-expressed MHC class I molecule on normal cells induces self-tolerance via the triggering of inhibitory receptors, namely killer immunoglobulin-like receptors (KIR) and the heterodimeric C-type lectin receptor (NKG2A) on NK cells [25]. Unlike T cells, NK cells lack an antigen-specific clonal T cell receptor and have the capacity to kill malignant cells without prior stimulation in an MHC-unrestricted manner. MHC mismatch, loss of MHC class I and/or overexpression of “stress ligands” on tumor cells upregulate the expression of activating receptors and thereby stimulate the cytolytic activity of NK cells via “missing-self” [25] and “stress-induced foreign” signals. Following target recognition, activated NK cells initiate cell death by activating tumor necrosis factor (TNF)-related death pathways (TRAIL, FasL), by releasing cytolytic granules containing apoptosis-inducing granzymes and perforin [26], or by complement- and/or antibody-dependent cell-mediated cytotoxicity (ADCC) via binding to Fc gamma receptors without prior antigen stimulation [27,28]. An optimal NK cell activity requires cell contact or cytokine-mediated signals from accessory immune cells in an inflamed microenvironment. The most effective cytokines produced by NK cells are interferon γ (IFN-γ), and granulocyte-macrophage colony-stimulating factor (GM-CSF), which can stimulate T helper-1 (TH-1) immune responses and the release of interleukins (IL)-2, IL-12, IL-15, IL-18, IL-21, interferon (IFN)-α, IFN-β, and IFN-γ from other immune cells that can activate NK cell-mediated cytotoxicity [29]. Considering immune cell memory, NK cells can re-encounter pathogens rapidly and robustly. These and other adaptive-like properties position NK cells at the border of innate and adaptive immunity [24].

Clinical concerns relating to the initiation of GVHD most likely mediated by allogeneic CAR T cells, and on-target off-tumor related toxicities have emerged [16,30]. CAR NK cells exhibit several efficiency, safety, and tolerability advantages over CAR T cells. CAR NK cell killing mechanisms can target heterogeneous malignancy by activating receptor-mediated signaling pathways which facilitate the elimination of cells that have lost or down-regulated expression of the antigen targeted by the CAR, thereby reducing the risk of relapse or resistance [13]. Despite their significant therapeutic potential at the peak of their expansion, CAR T cells can initiate severe inflammatory responses, including the CRS and ICANS resulting from an on-target off-tumor effect. These side effects can be life-threatening in some instances and require challenging post-treatment management [30]. In contrast, CAR NK cells produce lower amounts of pro-inflammatory cytokines such as IL-1, IL-6, TNF-α and, therefore harbor a lower risk for developing CRS and ICANS [15]. Although it has been shown that the low binding affinity of the CAR to its tumor target induces moderate pro-inflammatory cytokine secretion leading to mild neurological side effects, more clinical evidence is still required [31]. Prolonged survival of CAR T cells resulting in a long persistence in the body can exert negative side effects. Although long-term persistence of CD19 CAR T cells is important to achieve a durable anti-tumor immunity to prevent tumor recurrence, these cells can cause severe B cell aplasia, increased infectious complications [31,32], and cell fratricide due to the shared antigen expression on malignant and non-malignant T cells. An enhanced CAR T cell expansion and persistence may therefore act like a double-edged sword [33]. Moreover, the allogeneic “off-the-shelf” CAR T cell products can either induce or trigger GVHD indirectly, a clinical immuno-incompatibility syndrome which can lead to substantial morbidity and mortality, due to HLA mismatches between donor and recipient [34]. However, allogeneic NK cells exhibit a better safety profile which permits their use from healthy universal donors. The shorter lifespan of NK cells may reduce autoimmunity and long-term side effects [14,16]. Moreover, the inexpensive expansion makes both autologous and allogeneic NK cells ideal “off-the-shelf” candidates for a large-scale production of engineered CAR cells. Presently, NK cells of multiple sources such as peripheral blood (PB), umbilical cord blood (UCB), human embryonic stem cells (hESCs), induced pluripotent stem cells (iPSCs), and NK cell “like” lines (i.e., NK-92), are being tested in the context of CAR engineering (Figure 1) [8]. Despite relative low ex vivo expansion rates (2–4 fold) of mature NK cells which requires an apheresis to obtain adequate CAR NK cell counts (ranging from 1 × 10^7^ up to 2 × 10^9^ total CAR NK cells/kg body weight per infusion), feeder cell-based or feeder free expansion protocols enable expansion capacities more than 2000-fold in the case of UCBs, iPSCs and hESCs [35]. An unlimited expansion can be achieved in case of NK cell lines such as NK-92; however, for NK cell line-derived CAR NK cells, irradiation is required before ACT in patients. 

## 3. Precise Design for a Functional CAR Structure

The impact of CAR on immune cell function is dependent on the design and sequence of the utilized construct [20,36]. This offers the opportunity to rationally design CARs with desired features. In the following, we provide an overview on the sources of NK cells, CAR structure and the roles related to each element of the synthesized molecule in defining its efficacy. A classical CAR consists of (i) an ectodomain (extracellular antigen-recognition domain) linked to a hinge region, (ii) a transmembrane domain, and (iii) an endodomain (cytoplasmic signaling segment) (Figure 1).

The antigen-binding properties of a CAR depends on a single-chain fragment variant (scFv) which is mostly derived from a monoclonal antibody and mimics, almost entirely, the binding characteristics of the antibody from which it originates. In other words, the scFv plays a major role in specifically directing CAR cell affinity to a tumor cell surface antigen [37]. A fast receptor–ligand interaction due to the low affinity of scFv can affect the functionality of the CAR engineered cells. The CAT (a newly generated CD19 CAR) CAR T cells with a decreased affinity and a faster off-rate than the existing CD19 CAR FMC63 show higher anti-leukemic activity in preclinical models and patients [31]. Alternatively, an advanced scFv structure with bi-/multi-specificity has been designed to address antigen heterogeneity and to prevent tumor escape [38]. The variable heavy (VH) and variable light (VL) chain domains of an antibody are typically linked via a polypeptide (most commonly (Gly4Ser)3) to construct the scFv (Figure 1) [39]. The interactions between VH and VL shape scFv stability are influenced by the linker length and amino acid sequence. Since the chosen linker can mitigate the concerns of instability and inflexibility, ongoing research is mapping and designing new linker alternatives [40]. Fujiwara et al. showed that the meticulous configurations of the scFv framework altered the CAR structural stability, expression efficiency, and functional affinity [37]. The intrinsic instability of the scFv framework may promote CAR tonic signaling, a chronic and antigen-independent activation of CAR-equipped cells which leads to cell exhaustion and loss of activity [41].

The hinge region (also referred to as extracellular spacer) anchors the scFv to the transmembrane domain. It serves not only as a connector, but also as an influencer motif on the quality of CAR cell products [42]. The distance between the CAR scFv and target antigen required to form an optimal immune synapse is adjusted by the length-dependent flexibility of the hinge region [19]. For CAR NK cell engineering, the CAR spacers are often derived from the edited immunoglobulin G [43], CD8α [44], and CD28 [45]. The hinge domain may be a constructional factor controlling the clinical behavior of the CAR. This is partly attributed to the indirect footprint of the hinge and transmembrane domains which influence the level of inflammatory cytokine production and induce cell death capacity after antigen exposure [42]. The interaction with other immune cells is also driven by the hinge domain; for instance, via Fc-receptors when employing parts of immunoglobulins. Such interactions can either trigger CAR activation or lead to CAR cell depletion via the Fc-receptor engagement [46].

The transmembrane domain connects the CAR ectodomain to the intracellular signaling domain (Figure 1). The CAR containing CD8α and CD28-adapted transmembrane domain is frequently applied in primary NK cells, whereas CD28 is preferably used for NK cell lines. Alternative candidates such as 2B4 [47], HLA-A2 [48], CD3ζ, [49], NKP44, CD16, and NKG2D [50] have also been included into CARs designed for NK cells. It has been shown that hinge and transmembrane domains can alter the functionality of a CAR by regulating the threshold and amount of CAR signaling, respectively [51,52]. Moreover, they affect the expression levels of the CAR and thereby the antigen recognition [53].

The specific recognition and adequate stimulatory signals contribute to an ideal activation of the CAR NK cell. The CAR functionality can be regulated by intracellular signaling domains potentiated with a co-stimulatory helper signal (e.g., CD28 and 4-1BB) [51,54,55]. With respect to this, CARs are classified in different generations (Figure 1). The first generation of CAR NK cells consists of only a CD3 zeta (ζ) signaling moiety overcoming leukemia resistance to NK cell response [56]. One or multiple fragments have been included as co-stimulators to take advantage of diverse signaling pathways. The number of co-stimulators in the cytoplasmic domain determines the functionality, proliferation, and survival of cells [57]. However, side effects related to CAR T cells urge caution when applying this potent therapy in patients, especially for those constructs for which enhanced potency is expected [58]. The CAR domains for NK cell engineering are modeled from CAR structures optimized for T cells. To give an example, the 4-1BB co-stimulatory molecule which is meant to prevent exhaustion can increase the potency and persistence of CAR T cells to a larger extent than CD28 [59]. This difference can be attributed to various signaling pathways targeted by co-stimulators. The 4-1BB upregulates the phosphorylation of IKK α/β followed by stimulating nuclear factor κB (NF-κB)-induced apoptosis under the control of the tumor necrosis factor receptor-associated factor (TRAF) pathway [33]. Inserting 4-1BB into the chimeric CD19-CD3ζ receptor enhances the tumor-killing properties and kinetics of unmodified NK cells [55]. At the same time, ongoing research is aiming to identify a suitable (co-)stimulatory candidate which adapts to the unique characteristics of NK cells. In this regard, activating regulators such as DAP10, DAP12, or 2B4 have been productively employed to construct CAR endodomains [50,60]. 2B4, known as NK cell-specific co-stimulatory domain, has been shown to extend cytotoxic capacity via cytokine and contact-dependent mechanisms in vitro and in vivo [60]. A mesothelin CAR having NKG2D transmembrane and 2B4 co-stimulatory domains maximizes CAR NK cell anti-tumor activity, potentially because of possible downstream signaling pathways recruiting the endogenous DAP10 [50,61]. Despite the significant performance of the CD3ζ or DAP10 co-stimulatory domains when used alone, their combination amplifies the activatory signaling and subsequently NK cell-mediated killing towards hematological and solid malignancies [62]. The DAP12 adapter molecule has provided more remarkable anti-neoplastic potential than the CD3ζ signaling adapter, efficiently empowering CAR NK cells against prostate cancer stem cells. Consequently, based on signal activating potential, NK cell-specific co-stimulatory domains can be arranged in the order of DAP12 > CD3ζ > DAP10 [45]. Interestingly, a third generation of CAR NK cells containing DNAM1 and 2B4 has been shown to exhibit a much higher toxicity in hepatocellular carcinomas than CAR NK cells generated either with no co-stimulatory domain or with T cell-specific ones [63]. The upcoming fourth generation of armed CAR NK cells are engineered to co-express molecules such as cytokines with co-stimulatory domains to further improve functionality [8].

## 4. Established Targets for CAR NK Cells

Tumor eradication is often hampered by tumor heterogeneity modulating anti-cancer immune responses and the tumor microenvironment. The most common tumor escape mechanisms include impairment of antigen presentation, increased release of immunosuppressive factors, deficient cell death mechanisms, and increased damage repair which must be overcome to establish efficient therapeutic concepts [64,65]. 

The tumor-specific response is a fundamental principle of advanced cancer immunotherapy which can be positively triggered through the promotion of targeting tumor antigens (TAs) by immune effector cells. TAs undergo genetic/proteomic mutations and altered expression during oncogenesis, and are classified based on their expression pattern, predominantly into tumor-specific antigens (TSAs) and tumor-association antigens (TAAs) [66]. TSAs are derived from mutational processes in the cancer cells, and with a very few unique exceptions are not found on the surface of normal cells, whereas TAAs are expressed by cancer cells and normal cells, but are typically overexpressed in tumor cells [66]. In the absence of truly specific surface TSAs, TAAs are commonly used for targeted therapies, including immune cell-based approaches with CARs [64,66].

Various TAs have demonstrated promising activities in preclinical studies, and several of these have progressed into clinical trials (Table 1). Currently, CAR NK cells overwhelmingly use CAR constructs designed for T cells (Table 1). Their commonalities are their main moieties of the CAR structure (including scFv, transmembrane membrane, and signaling domain). The differences are mostly in the scFv affinity to target different tumor antigens or in the co-stimulatory domains developing different generations [14,15,16,17,18,19,20,21]. Since the majority of CAR NK cell studies are preclinical, insight into the efficiency and dependence of efficacy on the CAR structure is limited. However, there remains a need to optimize the structure of the CAR in terms of affinity and specificity for TA detection and the reduction of the incidence of complications which are attributed to on-target off-tumor effects. This latter complication is a major challenge in CAR T cell therapy mainly due to the expression of a common antigen in malignant and non-malignant normal cells of important organs [20]. In solid tumors, the choice of the CAR target is further complicated by physical or chemical barriers which prevent CARs from reaching their antigen in the tumor microenvironment [65]. In the pursuit of achieving high efficiency and restoring immune activity, various modalities to unleash cell activity or reduce barriers to efficiency are being considered. These include combination therapies with cytokine support, therapeutic checkpoint inhibitors, chemotherapeutics, oncolytic viruses, etc. [67]. Therefore, antigen selection is critical in CAR engineering to support therapeutic efficiency and safety. Here, we discuss the most common TAAs applied for CAR T cells and, recently, for CAR NK cell applications. 

### 4.1. CD19

Cluster of differentiation 19 (CD19) is a member of the immunoglobulin superfamily and a biomarker for normal and cancerous B cells and follicular DCs [14]. CD19 has been extensively studied as a promising therapeutic target for CAR-based immunotherapies due to its high expression density in B cell-derived neoplasms, including acute lymphoblastic leukemia (ALL), chronic lymphocytic leukemia (CLL), and non-Hodgkin’s lymphoma (NHL) [14,97,98]. CD19 CAR T cells against various B cell malignancies have been approved by the FDA and European Medicines Agency (EMA) [99]. Despite the outstanding efficiency of targeting CD19, experience with monospecific CD19 CAR T cells has revealed resistance or tumor escape associated with a loss or diminished antigen density under therapeutic pressure in 30–70% patients [12,30]. The potential resistance mechanisms to CD19 CAR therapies include CD19 gene mutation or downregulation, cancer cell selection by an immune response, lineage transdifferentiation, or fratricide killing of CAR T cells as a result of trogocytosis [100]. Interestingly, delayed CAR expression on T cells attenuating the tonic CAR signaling can decrease the fratricide event in T cell malignancies. The transient control of CAR expression could be obtained upon retroviral transduction by a Tet-OFF system. In this system, the presence of doxycycline (DOX) during in vitro culture prevents transactivator (rTA) binding to the synthetic promotor resulting in minimal CAR expression. By removing DOX prior to injection into mice, T cells could acquire sufficient CAR expression and increased survival rates [33]. Dual or multi-targeted CAR manufacturing can be a solution for patients with tumor resistance and relapsed/refractory cancers. Several studies have shown the feasibility and potency of simultaneous CAR targeting of CD19 along with other alternative antigens, such as CD22 [101] and CD123 in AML [102]. It has been shown that CD19-positive B cells infiltrate into the pancreatic ductal adenocarcinoma and may limit the CAR T cell activity. According to these findings, mesothelin and CD19 CAR T cells were administered to three patients in order to target tumor cells and to deplete B cells. Although the stable disease was the best achievement of this dual therapy, both its safety and efficiency need further improvement [103]. Trivalent CARs targeting CD19, CD20, and CD22 on tumor cells can also be used to treat CD19-negative, relapsed tumors [104]. In a first clinical trial using a CAR NK cell-based adoptive therapy [14], the side effects associated with CAR T cell therapies such as cytokine-release storm and neurological symptoms have not been observed, and this experience supports further investigations of this approach [14]. The first clinical results of CAR NK cells are encouraging and justify further assessment of CAR NK cells across different indications [98,105,106]; however, efficacy needs to be compared to that achieved with conventional CAR T cell concepts.

### 4.2. EGFR

Epidermal growth factor receptor (EGFR; HER1; ErbB-1) belongs to a four-member transmembrane receptor (ErbB) family with tyrosine kinase activity. An abnormal alteration in EGFR expression and activation can transform a healthy cell into a cancerous cell, and this is a hallmark of many types of epithelial carcinomas such as breast, lung, renal, or head and neck cancer and glioblastoma [107]. Thus, a suitable therapeutic strategy can be built upon a comprehensive understanding of the mechanism of action of EGFR and its impact on tumor development. Specific receptor blockers which inhibit the EGFR-induced signaling pathway can prevent cancer progression [108]. In addition, mutation-derived resistance toward receptor blockers forces precision medicine to identify a more specific targeted therapy. In the last decades, EGFR CAR cells have shed light into the potential of CAR-based adoptive immunotherapy [68,109]. Combining EGFR CAR NK cells with oncolytic herpes simplex virus-1 or chemotherapeutics showed synergistic effects on breast cancer and renal cell carcinoma (RCC) in mice, respectively [110,111]. Concerning the positive effect of IL-15 on NK cell persistence, tumor infiltration, and functionality, a herpes simplex-1-based oncolytic virus expressing IL-15/IL-15Rα has resulted in highly significant anti-glioblastoma effects in combination with EGFR CAR NK cells [112]. A mutant form of EFGR (named EGFRvIII) expressed on glioblastoma cells is one of the few potentially cancer-specific mutations that can be found on the cell surface of glioblastoma cells and is thus amenable to CAR NK [54,113] and CAR T cell treatment. The co-culture of glioblastoma cell lines with the EGFRvIII CAR NK cells has been shown to induce efficient and specific apoptosis in cancer cells. For this approach, the human NK cell line KHYG-1 was successfully transduced with lentiviral vectors and then sorted to reach an efficiency above 80% [114]. Combination therapies including CAR cell therapies have attracted an increased interest to overcome clinical tumor recurrence. To overcome the impaired colony-selection and presence of residual cancer cells, dual CAR cell targeting strategy of EGFR with other target candidates could promote the efficacy of the treatment. Notably, dual targeting is achieved by co-expressing two specific CARs or expressing a bispecific CAR (a shared epitope for EGFR and its mutants) on a single immune cell platform [38,54,115]. As an alternative, the blockage of immune regulatory checkpoints of T cells via CRISPR/Cas9-mediated gene editing could also improve the outcome of CAR cell therapies [116]. Similarly, inhibiting immunosuppressive genes that are activated upon IFN-γ secretion by EGFR CAR T cells can improve the efficacy of targeted therapies, in vivo [117]. Ongoing efforts are attempting to apply EGFR CAR cells in clinical trials.

### 4.3. HER2

The human epidermal growth factor receptor-2 (HER2; ErbB-2) is an oncogenic tyrosine kinase (also termed as HER2/neu) of the ErbB family. What makes HER2 superior for a targeted cancer therapy is the differential expression pattern in normal and neoplastic cells. Given the vital role of HER2 in tumorigenesis, the overexpression of HER2 attributes multiple features of malignant characteristics to certain solid tumor types such as breast cancer and glioblastoma [118]. Hence, the HER2 expression level can be considered as a target for cancer prognosis and diagnosis. HER2-based treatment strategies involve either the direct targeting of HER2 or related signaling pathways [118,119]. However, an aberrant form of HER2, known as p95, which has no extracellular domain, leads to therapeutic resistance and highlights the urgent need for more effective therapies [120]. Although changes in HER2 expression levels during cancer pathogenesis have raised concern for targeted therapies, it still attracts considerable attention as a target for designing CAR T cells, and recently also for CAR NK cells [121]. Long-term persistence in vivo and efficient cell killing strongly favors CAR-directed cells for clinical use over approved targeted immunotherapies using HER2-specific antibodies (e.g., Trastuzumab) [51,122,123]. Despite the value of well-established CAR T cell approaches, CAR NK cells have been shown to be potent against solid tumors and show a good safety profile [124]. As an example, the successful performance of HER2 CAR NK cells in primary and immortalized cell types has been validated by marked and selective anti-tumor activity against cancers like rhabdomyosarcoma, in vitro and in vivo [51,125]. The promising effectiveness of the CAR cell therapy associates with an optimized HER2 CAR design which in turn depends on scFv affinity and the level of HER2 amplification in carcinomas. As another strategy, the clinical applicability of CAR-modified cells can be potentiated by utilizing dual targeting and inhibitory CARs which limit off-target effects [126]. Local treatment with combinatorial adenovirus vector with simultaneous oncolytic and checkpoint inhibition ability as well as expressing pro-inflammatory cytokines have enhanced the activity of systemically administered HER2 CAR T cells against xenograft and orthotopic head and neck squamous cell carcinomas (HNSCC) in mice [127]. Although HER2 has been one of the most pioneering TAAs that have been used for designing CAR constructs, more work is required in order to achieve sustained clinical benefits with lower safety concerns.

### 4.4. EpCAM (CD326)

The epithelial cell adhesion molecule (EpCAM, CD326) is expressed by most epithelial tissues, and its expression is up-regulated on certain tumor types, including adenocarcinomas and squamous cell carcinomas [128]. EpCAM is thought to have a critical function in cancer and to drive malignant properties of tumor cells. For instance, the presence of EpCAM contributes to resistance to chemo/radiotherapy in prostate cancer. In addition, it has also been known as a circulating biomarker of metastatic tumors. Hence, this TAA has been extensively studied in the context of cancer diagnostics, prognostics, and therapeutics [129]. The selection of EpCAM as a candidate for a targeted therapy originates from its uniform cell surface distribution on cancer cells which is different from the basolateral localization of EpCAM in normal cells. EpCAM-targeted treatment approaches are mainly based on monoclonal antibodies. Vaccines based on EpCAM targeting have shown a low efficiency in the metastatic state, and to induce acquired resistance and immunogenicity [130]. In contrast, the therapeutic results obtained with EpCAM CAR-directed cells in models of solid ovarian, colon, lung, and breast tumors are promising [131,132]. EpCAM CAR NK-92 cells combined with a tumor kinase inhibitor (e.g., Regorafenib) achieved significant anti-tumor responses in human colorectal cancer xenografts [70]. Notably, the expression of EpCAM in normal tissues resulted in dose-dependent severe side effects and even death upon the infusion of EpCAM CAR T cells into BALB/c mice in a colon tumor model [133]. Taken together, although EpCAM CAR-directed cells offer therapeutic promise, safety considerations are of the utmost importance.

### 4.5. GD2

The disialoganglioside (GD2), unlike other gangliosides, is thought to be an excellent target for immunotherapy of different solid malignancies (e.g., neuroectoderm-derived neoplasms, most melanomas and Ewing sarcomas) due to its reduced expression in healthy cells [134]. Although the FDA-approved GD2 antibody (Dinutuximab) exerts anti-tumoral activity via antibody-dependent cell- and complement-dependent cytotoxicity in high-risk neuroblastoma, the therapy is accompanied by significant treatment-related adverse effects [135]. Due to the recurrence of neuroblastoma, the Dinutuximab monotherapy has been tested as part of a combined modality consisting of Dinutuximab and ex vivo activated human NK cells. The combination showed a synergistic killing activity as well as a suppressive effect on the invasiveness potential of three different neuroblastoma cell lines in vitro. Indeed, after resection of neuroblastoma, this strategy could improve the overall survival in an immunodeficient mouse model by limiting the aggressiveness of residual tumor cells [136]. These results have led to the use of GD2 CAR T cells and CAR NK cells for treating breast cancer and neuroblastoma, respectively [137,138,139]. The inhibition of immunosuppressors such as IDO1 (indoleamine-pyrrole 2,3-dioxygenase1) in the microenvironment of neuroblastoma has the potential to limit tumor escape and to enhance the synergistic anti-tumor effect of GD2 CAR engineered T and NK cells. This novel immunotherapy concept has entered phase I/II clinical trials [140]. Moreover, it has been shown that co-expressing IL-15 and GD2 CAR potentiated the direct anti-neuroblastoma effect of Vα24-invariant natural killer T (NKT) cells. Interestingly, given the long-term in vivo persistence, satisfying tumor trafficking, and no significant toxic adverse effect of a GD2-CAR-IL-15 NKT, a first-in-human clinical trial has been performed in children with relapsed or resistant neuroblastoma following pre-treatment with Cyclophosphamide/Fludarabine (Cy/Flu). According to the International Neuroblastoma Response criteria (INRC), the therapy achieved stable disease in two patients and partial remission in one patient [139,141]. Overall, GD2 has the potential to be established as a biomarker for targeted therapy in certain tumor types, especially neuroblastoma. 

### 4.6. Mesothelin

Mesothelin (MSLN) is a cell surface adhesion molecule which is overexpressed in 20–90% of cancer entities such as mesothelioma, triple negative breast, ovarian, lung, and pancreatic cancers. Low expression rates are detected in non-critical normal tissues. Hence, MSLN might be a better choice for CAR targeted therapy of solid tumors owing to the potentially lower risk of off-target effects [142]. Furthermore, MSLN has been found to be expressed on AML cells but not on normal hematopoietic cells [143]. A large number of preclinical and clinical studies have considered MSLN as a basis for targeted therapies using antibodies, immunotoxins, vaccines, and CAR-engineered cells [144]. NK-92 cells armed with a MSLN CAR have been reported to perform well in gastric cancer in vitro and have shown efficacy in vivo with minor CAR-mediated toxicities [145]. Similar positive results have been observed in MSLN-positive ovarian cancer cell lines treated with MSLN CAR NK-92 cells. In addition, these CAR NK cells can specifically target intraperitoneal ovarian tumors and enhance survival of tumor-bearing mice [72]. Achieving more accurate and convincing MSLN CAR cell-mediated targeting is the primary goal of several research groups but requires the tackling of various challenges such as production, infiltration into solid tumors, and overcoming the immunosuppressive effects of the tumor microenvironment [146,147]. To achieve this, some researchers use MSLN CAR NK cells not only as specialized tumor cell killers, but also as carriers to deliver inhibitors or drugs to the tumor microenvironment to overcome resistance and immunosuppression [148]. Like other TAAs, the dual CAR targeting system has delivered successful therapeutic outcomes [103]. Similarly, interfering with immunosuppressive interactions such as PD-1/PD-L1 via antibodies or drugs can promote the potency of CAR cells [149]. Although the MSLN CAR-directed cells need to be optimized to address many issues such as safety, MSLN is a prominent TAA for targeting therapies, particularly in solid tumors. 

### 4.7. HSP70

The major stress-inducible Heat shock protein 70 (Hsp70) is a cytosolic protein which resides in nearly all nucleated cells. Due to their heightened growth rates and energy demand [150], most solid tumor cells as well as hematological malignancies show an up-regulated expression of Hsp70 in the cytosol. Moreover, tumor cells, in contrast to normal cells, present Hsp70 on their plasma membrane [151]. This tumor-specific Hsp70 membrane expression is enabled by globotriaoslyceramide (Gb3) which is not present on the cell membrane of normal cells [152]. Our group has developed a monoclonal antibody termed cmHsp70.1 which is able to detect membrane bound Hsp70 on viable tumor cells [153]. Stress, including radio- and chemotherapy, increases the cell surface density of Hsp70, and the density is also greater in aggressive tumor cells and metastases [154]. This unique membrane form of Hsp70 might therefore qualify as an excellent target for cell-based immunotherapies [155]. A stimulation of NK cells with an Hsp70-derived 14-mer peptide TKD and IL-2 has been shown to increase the cytolytic and migratory capacity of NK cells against membrane Hsp70-positive tumor cells [156]. The tolerability and efficacy of Hsp70 pre-activated NK cells in recognizing and killing of membrane Hsp70-positive tumor cells has been demonstrated in preclinical settings [157], a pilot study [6,158], a phase I clinical trial in patients with metastatic colorectal cancer or NSCLC [159], and a phase II clinical randomized trial in patients with advanced NSCLC [5]. Considering the findings of these studies, it can be postulated that genetically engineered CAR T/NK cells with a single chain cmHsp70.1 Fv fragment might provide a promising strategy to treat tumor patients with highly aggressive and therapy-resistant tumors expressing membrane Hsp70. 

## 5. CAR Transfer Methodology into NK Cells

The successful production of genetically engineered cells relies on highly efficient and safe genetic delivery systems. In this respect, modern biotechnology has provided the capability to design diverse viral or non-viral platforms for CAR delivery [160,161]. Emphasis is towards gene editing and delivery systems that achieve stable and targeted CAR integration into the genome and yield therapeutically qualified CAR NK cells which can be incorporated into the clinical workflow [21] (Figure 2). Here, we provide an overview of the currently used methodologies that are applied for CAR cell engineering, with a focus on NK cells.

### 5.1. Viral Vectors for CAR Transduction

Over the years, viral vectors have offered ideal vehicles for introducing genetic material into cells for the purpose of preclinical and clinical gene therapy [162]. A great spectrum of viral vectors has been investigated for gene delivery into immune cells, among which retrovirus and lentivirus-based vectors are commonly used as CAR carriers as they have several advantages such as reliable packaging size of >8 kb, stable genome modification by integration into the host genome, and long-term maintenance of transgene expression in the host [162,163].

The retroviridae family has been extensively used as an efficient gene delivery vector to various types of mammalian cells, and most recently for CAR transduction of NK cells [163]. Only a single dose of retroviral vectors has been shown to achieve the required CAR transduction efficiency in human pre-stimulated NK cells [164]. Clinically, 7 of 11 patients reported complete remission of lymphoid tumors (non-Hodgkin’s lymphoma and CLL) in response to retroviral-transduced CAR NK cell therapy, mostly in combination with other therapies. After a one-year follow-up, the detection of CAR NK cells in the peripheral blood of treated patients indicated long term-persistence of retrovirally CAR-modified NK cells [14]. Amongst different generations of retroviruses, alpha-retroviral vectors have displayed superiority over gamma-retroviral vectors, with transduction effectiveness of 90% in the NK cell line and >60% in stimulated primary NK cells [165]. However, the unique replication ability of retroviruses allows stable transgene integration into the target genome and long-term CAR expression, but only in dividing NK cells [164]. In addition, retroviral vectors have a high mutagenesis rate and uncontrolled immunoreactions in the recipient which are attributed to their random insertion into the genome resulting in unnecessary high titers and sustaining persistence of viral vectors [166]. It is important to bear in mind that these potential drawbacks have not shown to be disadvantageous clinically when applying CAR T cell therapy; in fact, one of the approved CAR T cell products employs a retroviral vector. 

Lentiviruses, a subcategory of the retrovirus family, are genetically more complex than most retroviruses due to various encoded regulatory and accessory proteins. Although retrovirus-mediated gene delivery is common, the noteworthy clinical prosperity of lentiviral CAR-transduced cells is increasing [167]. Lentiviruses have no dependency on cell cycle progression and could be successfully transduced into both cycling and non-cycling NK cells [168]. However, primary cultured NK cells usually require multiple rounds of transduction by viral vectors to achieve an adequate transduction efficiency [168,169]. The semi-random integration of lentiviral vectors still carries the risk of insertional mutagenesis and dysregulation, but to a lesser extent than retroviral vectors and may therefore provide a safer alternative. This difference has been attributed to differences in the degree and integration site selection of lentiviral and retroviral vectors [166]. Since the integration site distribution can affect the therapeutic outcome of lentiviral-transduced CAR T cells, a targeted knock-in of transgenes can be used to improve the efficiency [170]. Moreover, the safety concerns relating to the use of lentiviral vectors have been bypassed by advances in their design [167]. 

Since the use of viral vectors faces many challenges due to their limited potential to deliver genes into primary NK or T cells, scientific attempts to overcome the barriers are of great importance. In this regard, there are several possible optimization approaches, such as NK cell stimulation with feeder cells, using cytokine cocktails to accelerate the uptake of viral vectors, or manipulating the innate antiviral response of NK cells. The fundamental aim of these approaches is to resolve the resistance to transfection or infection using routine methods [51,168,171]. Additionally, the cell tropism and host range of the viral vectors can be altered by substituting native envelope proteins with heterogeneous proteins, most commonly the G glycoprotein of vesicular stomatitis virus (VSV-G), in a procedure called pseudotyping [172]. The VSV-G-based viral vectors provide high potential for a stable CAR transduction in T cells, but have been less successful in primary NK cells due to an induced genotoxicity [173,174]. It has been recently reported that upregulation of low-density lipoprotein-receptor (LDLR) expression, which is triggered by VSV, can boost the VSV-G lentiviral transduction into primary NK cells [175]. As an alternative, the baboon envelope glycoprotein (BaEV-gp) along with cytokine treatment has induced a higher affinity of lentiviral vectors to NK cells [176,177]. Similarly, pseudotyping with envelop protein of feline endogenous retrovirus envelopes (RD114) has been shown to increase the infectivity level of viral vectors in NK cells, with no significant changes in cell viability [106,165,178,179]. By considering the positive correlation between the virus diffusion on target cells and subsequent infection, cationic polymers such as hexadimethrine bromide (polybrene) or protamine sulfate can neutralize the negative charge of some enveloped viral vectors, thereby facilitating their membrane fusion [180]. Moreover, transduction enhancers such as retronectin or vectofusion-1 promote the adhesion of pseudotyped viral vectors to the host cell surface, which in turn increases cellular internalization. The greater transduction efficiency results in a stronger anti-leukemia capacity of NK cells. Up to 90% of CD19 CAR transduction in PB-derived NK cells has been achieved using retroviral vectors pseudotyped with RD114-TR in the presence of retronectin [106,165]. Compared to T cells, NK cells are more resistant to viral transduction guided by their pattern recognition receptor signals [181]. Hence, inhibiting intracellular antiviral defense mechanisms can enhance the entry of VSV-G pseudotyped lentivirus vectors into primary NK cells and more prominently into immortalized NK cells [171].

### 5.2. Non-Viral Techniques for CAR Transfection

Although viral vector-mediated gene delivery is currently the dominant strategy for most CAR NK cell engineering systems, this approach still deserves much attention, particularly regarding safety issues. Therefore, non-viral systems have received attention for a number of reasons including safety, outstanding design flexibility, and simplicity of large-scale production of therapeutic cells at a reasonable pace and low cost [182]. In the following, we review the most popular methods applied for CAR cell engineering which are frequently based on membrane permeabilization and carrier-based gene transfer.

#### 5.2.1. Electroporation-Mediated Delivery

Electroporation is a non-viral technique for the loading of naked nucleic acids into cells which involves temporary permeabilization of the cellular membrane. Notwithstanding the acceptable electrotransfection efficiency of CAR-encoded DNA [183,184], several reports have suggested risks of growth arrest and death of NK cells hindering DNA application [185,186]. In a comparative study, DNA-electroporated NK-92 cells showed two to three times lower cell viability than mRNA-transfected NK-92 cells after 24 h [185]. Therefore, a greater deal of attention has been dedicated to mRNA electroporation because of the high transfection efficacy of NK cells and a better cell survival profile [185,187,188]. The electroporation of mRNA encoding a CD19 CAR in a single-step procedure mediated high expression in primary unstimulated and expanded NK cells with no significant impairment in cell viability. The transfected NK cells became strongly cytotoxic against CD19-positive leukemic cells, to a lesser extent in primary than expanded NK cells, thereby highlighting the positive correlation between transfection efficacy and killing ability of NK cells [189]. In accordance with results obtained from primary NK cells, immortalized cell lines such as NK-92 electroporated with CD19 CAR mRNA yielded a 10-fold greater transfection efficiency than with cDNA, and this translated to higher lytic activity towards resistant CD19-positive chronic lymphocyte leukemia cells [185]. The NK-92 cell line exhibited an efficient expression of the CD20 CAR following both mRNA electroporation and lentiviral transduction. Conversely, the CAR transfer to UCB-derived NK cells with mRNA was not as efficient as lentiviral-mediated transduction [190]. Given the findings of such investigations, one can state that lentiviral vectors are still promising with respect to achieving sufficient transduction rates in primary NK cells. In this case, the advantage of rapid mRNA translation to CAR cannot be neglected. Since mRNA works as a transcript replacement in the cytoplasm, the transfection rate attained up to 80% efficiency within several hours after mRNA electroporation, whereas the level of receptor expression peaked at least one week after retroviral transduction [191]. 

Although there is sufficient evidence to conclude that mRNA electroporation is a simple, fast, and cost-effective manner, the half-life of the induced transient CAR and the progressive decline in its expression influences the therapeutic time window [191]. However, unstable CAR expression and short-term interactions with tumor target cells have not impaired the potential of CAR NK cells. After the electroporation of a CD19-BB-ζ CAR into expanded NK cells, the CAR expression reached a maximum during the first 48 h. Interestingly, significant cell killing of ALL cell lines was apparent at the same time. Furthermore, despite the 4-day persistency of the CAR NK cells, satisfying anti-leukemia effects were also observed in vivo [191]. Nevertheless, in some cases, multiple injections of mRNA CAR NK cells are required to affect and delay tumor growth [186,189]. Albeit nonsurprisingly, multiple administrations of CAR NK cells may also induce undesirable immune responses. mRNA electroporation is an adaptable and reliable way to meet the demand of large-scale manufacturing of CAR NK cells for clinical trials and may overcome the concern of on-target off-tumor toxicity against healthy tissues after infusion [192]. 

Since immune cell engineering using electroporation-based methods has significant potential, substantial efforts have been made to establish an optimal electrotransfection protocol with minimal damage of normal cells [191,193,194]. Nucleofection is an electroporation-based system utilizing cell-type-specific buffer solutions and optimized electric settings for rapid and efficient DNA or mRNA delivery to a wide variety of cells such as stem cells and primary cells that are resistant to conventional gene transfection methods. This method can direct gene delivery into the nucleus without dependency on cell division [195,196,197]. The applicability of this transfection mode has been tested in NK cells. According to the results, nucleofection with CD20 CAR mRNA result in an enhanced anti-tumor activity against CD20-positive hematologic malignant cells in cell culture as well as in animal models [80]. The co-transfection of a CAR sequence with other therapeutic nucleic acids provides a promising electroporation or nucleofection approach which might be able to obtain optimal CAR-engineered cells [198].

#### 5.2.2. Microfluidics-Based Cell Squeezing

Cell squeezing devices based on the microfluidic technologies have been rapidly developed for an efficient cytosolic delivery of various macro- and nanomolecules into different cell types. The principle of this method is based on a mechanical modification of cells that allows a transient permeabilization of cellular membrane [199]. Unlike electroporation, the microfluidic cell squeezing has a minimal adverse effect on immune cell functionality, gene transcription profile, and cytokine release, all of which are critical issues to be considered [32]. However, the detailed impact of electroporation and microfluidic cell squeezing on immune cell phenotype and innate functionality (e.g., NK cells) needs to be further elucidated. This approach is expected to become valuable as a virus-free delivery strategy in the near future.

#### 5.2.3. Nanocarrier-Mediated Delivery

Chemically programmed vectors have been considered as a turning point in the development of non-viral gene delivery systems [200,201]. The goal is to supply enhanced intercellular penetration followed by sufficient endosomal escape and specific nuclear or cytosolic localization of the cargo to bypass biological barriers [202]. The most promising synthetic vectors include cationic polymers, lipids, or combinations of them to achieve safety, stability, or efficiency outcomes. However, these approaches still need to be optimized [203].

The advantages offered by nanotechnology open the possibility to implement nanoformulations in the CAR cell technology. Among many other options, the transfection competency of CARs encoding mRNA or DNA by polymeric nanoparticles showed a high compatibility with T and NK cells [204,205]. Compared to electroporation, the use of mRNA-carrying nanoparticles has shown a higher viability and higher expansion rates of T cells [206,207]. Synthetic core-shell particles complexed with pDNA EGFR CAR have been shown to efficiently transfect the NK-92MI cell line in a dose-dependent manner. The expression of EGFR CARs potentiated the anti-oncogenic activity of NK cells in a xenograft breast tumor model [208]. Interestingly, the potency of adoptive therapy can be monitored by optical imaging of CAR NK cells transfected with fluorescent-labeled polymeric particles [208].

Ionizable lipid nanoparticle (LNP) platforms are more attractive than polymeric nanoparticles, as they allow stable formulation, potent endogenous cellular internalization, and low toxicity rates [209]. The ease of LNP modifications led to optimized-performing formulations for mRNA CAR transfer to T cells [207]. A novel charge-altering releasable transporter composed of lipophilic and polycationic blocks has been reported to significantly facilitate the delivery of an mRNA-based CAR into resting NK cells without any need for pre-activation. This procedure was advantageous over an electroporation method [205].

Receptor targeting plays a vital role in smart gene delivery by non-viral and viral vectors, especially in animal models [210,211]. CD3-targeted nanoparticles encapsulating a CD19 CAR gene flanked with a piggyBac transposon has been shown to deliver a robust CAR production in dividing T cells in vitro and circulating T cells in vivo [212]. Insights drawn from studies have proven that precise polymers assembled in a sequence-defined manner can also be ideal for various cargo deliveries [213]. The therapeutic nucleotide codes transferred by nanovectors are commonly used in the forms of pDNA and mRNA [214]. However, the stronger and less cell-cycle dependent delivery makes minicircle DNA a better choice than plasmid DNA for complexation [215]. 

The flexible design and transfer capability for versatile types of cargos, either alone or in combination, make the nanoscale delivery system a key method in biotechnology and, most recently, in the CAR cell engineering field [212]. Nanoparticles have become increasingly popular platforms due to their ability to carry transposon for a non-viral, stable CAR transfection in combination with CRISPR/Cas9 for a targeted gene integration.

#### 5.2.4. Transposon System

The application of transposon-based vectors has attracted much attention among other non-viral transfection systems for CAR engineering, satisfying requirements for sufficient efficiency, permanent transgene expression, together with low immunogenicity and no genotoxic effects. Notably, the production of plasmid-based vectors is time- and cost-effective. Due to these advantages, the transfection potential for large genes (more than 100 kb) makes this system a valuable method for cell engineering [216,217,218]. It typically consists of mobile plasmids (transposons) flanked by two terminal inverted repeats (TIRs) carrying the enzymatic gene and the inserted sequence of interest. The transposase mediates cutting-and-pasting of the desired elements in the host genome [219]. Common transposons such as piggyBac and sleeping beauty have led to favorable therapeutic cell manufacturing, of which sleeping beauty-based CAR-modified cells are currently being tested clinically because of their potential and safer integration profile [50,220,221,222]. Using this approach, CAR integration by sleeping beauty-mediated transposition remarkably increased the killing capacity of NK-92MI cells toward pancreatic cancer cells in vitro [146]. Li et al. published that the strengthened cytolysis of CAR- iPSC-derived NK cells against resistant ovarian tumor cells raised from the efficient transduction potential of non-viral piggyBac transposon vectors [50]. Despite the self-regulation hypothesis, transposase over activity is a hard-to-control phenomenon which can cause cytotoxicity, transgene remobilization, and undesirably malignant transformation of the therapeutic products [223]. To gain precise control of the transposon-mediated gene engineering with satisfactory safety and efficiency, conventional DNA plasmids have been replaced by a combination of short-lived mRNA encoding transposase and minicircle DNA cargo [146,224]. The mRNA-encoded transposase can be applied for hyper controllability of transposition events in the system [224]. Co-electrotransfection of minicircle DNA containing a transposon coding for a CD19 CAR and transposase enzyme can lead to a favorable insertion profile and generation of CAR T cells with a high therapeutic potential [225]. What may also limit the application of the transposon is the hurdle of delivery. Although transfer with viral vectors and electrotransfection could experience rapid development to get into clinical trials, further attempts still need to improve transposon delivery [217,221]. For instance, polymeric nanomicelles have recently shown a safe and site-specific delivery of piggyBac transposons into T cells via the endocytotic pathway. These reduction-sensitive particles could form stable complexations with CAR transposon and transposase plasmids at an optimized N/P ratio and resulted in an acceptable transfection outcome. However, the cell number, plasmid concentration and the T cell cycle can alter the transfection efficiency [226]. In addition to the efficiency, the specificity, and safety of CAR gene transfection using the transposon machinery can be optimized by considering different factors related to the transposon system (e.g., plasmid origin and transposition activity), as well as the target cell type. Moreover, the delivery platform and the culture condition affect the system potency [146,222]. Nevertheless, the translation of transposon technology into medical application must be carefully monitored. Most recently, CAR T cell lymphoma has been observed in two of ten patients effectively treated with piggyBac modified CD19 CAR T cells [227].

#### 5.2.5. CRISPR-Cas9

CRISPR-Cas9 is being used as a gene editing tool to ameliorate the efficiency of CAR cell products [228,229] Based on this technology, a programmable single guide RNA (sgRNA) brings a Cas9 nuclease to a specific site of the genome for inducing double-strand breaks followed by integration of the desired gene cassette via endogenous DNA repair mechanisms, homologous recombination (HR) or non-homologous end-joining (NHEJ). The productivity of CRISPR-Cas9 depends on the applied format of CRISPR-Cas9 components, transfection method, and cell type [230]. The CRISPR-Cas9 elements can be introduced into cells in multiple formats such as DNA, RNA, and ribonucleoproteins (RNPs). Accordingly, the delivery mode varies according to the CRISPR-Cas9 component option [231]. 

To date, NK cells have shown resistance against transfection methods which highlights critical concerns for gene-editing-based immunotherapy with CRISPR-Cas9. Up to now, many strategies have been exploited in an effort to establish a safe and efficient delivery platform [231,232]. The targeted CAR integration into the T or NK cell genome prevents aberrant viral unspecific and semi-randomly gene integration. CD19 CAR knock-in into the T cell receptor α constant (TRAC) locus ensures a better performance over the conventional CAR T cells against acute lymphoblastic leukemia models, as a consequence of effective internalization and a more uniform expression of the CAR [233]. However, the outcome is influenced by the delivery efficacy of CRISPR-Cas9 components as well as the gene of interest. The therapeutic qualification of CAR-engineered immune cells can be improved via CRISPR-Cas9-mediated editing of major inhibitory genes such as PD-1. In view of the fact that the programmed cell death ligand 1 (PD-L1) expressed on tumor cells negatively acts on cancer treatment and can suppress the CAR T cell function [234], the blockade of PD-1 receptor by Cas9-nucleofection accompanied by lentiviral CAR transduction could enhance the anti-tumor activity of T cells [116]. Comparably, targeting the cytokine checkpoint by Cas9 accelerated the therapeutic role of CAR NK cells. A fourth generation of CAR NK cells transduced with a CD19 CAR and IL-15 has been shown to promote in vivo persistency and subsequent efficacy against lymphoma which were potentiated after CISH (cytokine-inducible Src homology 2 domain) gene knockout [235]. All in all, the scientific evidence has declared the proof-of-principle for Cas9-based gene editing, which may expand the landscape of CAR cell engineering in future approaches.

## 6. Conclusions

Genetically engineered CAR NK cells are a novel and very promising cell therapy approach in oncology. Since clinical application is still in its infancy, optimization strategies are being pursued to bring a safe and robust anti-tumoral product into clinical practice. 

## Figures and Tables

**Figure 1 cells-10-03390-f001:**
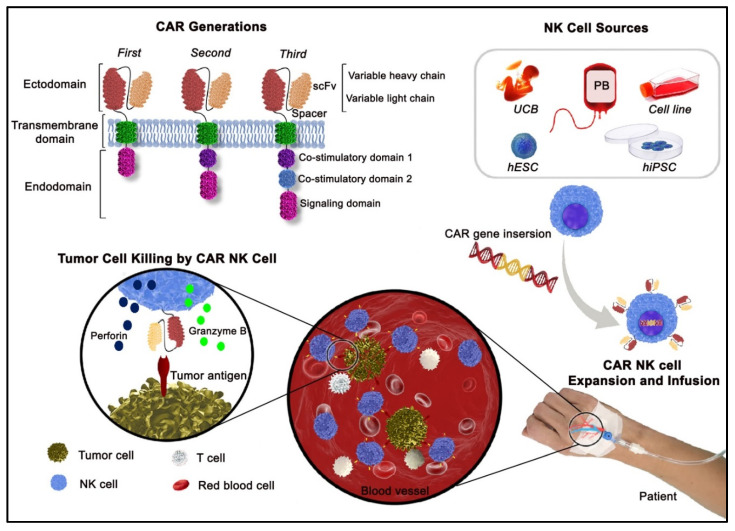
Schematic illustration of the first, second and third generations of chimeric antigen receptors (CARs), sources of natural killer (NK) cells for genetic modifications with CARs, and CAR NK cell-based immunotherapy. Abbreviations: PB, peripheral blood; UCB, umbilical cord blood; hESC, human embryonic stem cell; hiPSC, human induced pluripotent stem cell.

**Figure 2 cells-10-03390-f002:**
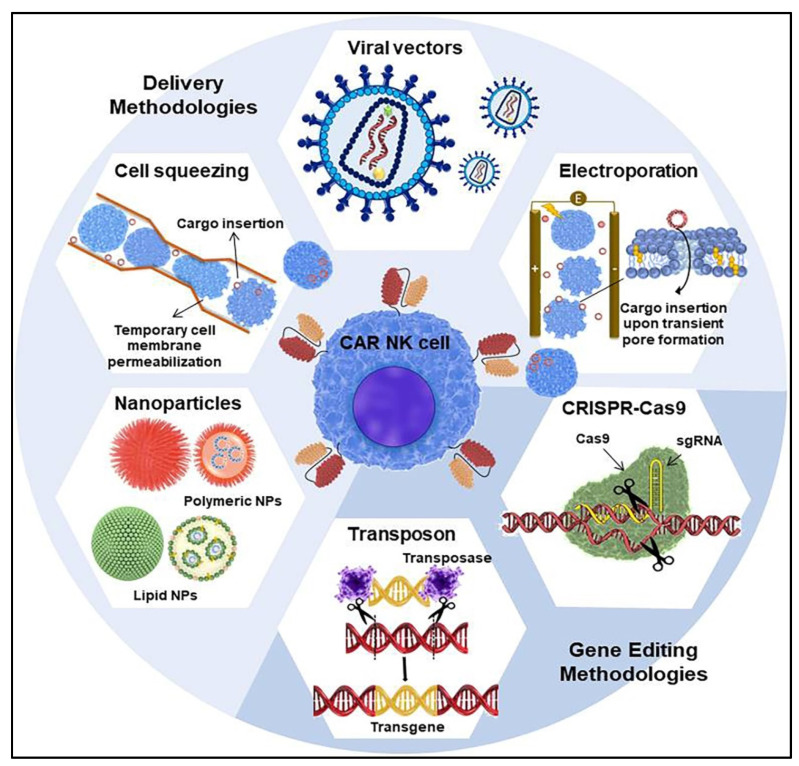
A schematic illustration of the viral and non-viral methods for CAR structure delivery to NK cells.

**Table 1 cells-10-03390-t001:** Overview of the most common tumor-associated antigens (TAAs) targeted by CAR NK cells in preclinical and clinical studies (the references refer to the most recent published study for each TAA). Abbreviations: CB, cord blood; PB, peripheral blood; iC9, inducible caspase 9; TF, tissue factor.

Tumor Antigen	Tumor	Source	Delivery Method	Transmembrane + Endodomain	State	Ref
CD19	Non-Hodgkin’s lymphoma Chronic lymphocytic leukemia	CB NK	Retroviral	CD28, CD3ζ, iC9, IL-15	Phase I and II trial	[14]
EGFR	Triple-negative breast cancer	PB NK	Lentiviral	CD8, CD28, 4-1BB, CD3ζ	Preclinical	[68]
HER2	Glioblastoma	NK-92	Lentiviral	CD28, CD3ζ	Preclinical	[69]
EpCAM	Colorectal cancer	NK-92	Lentiviral	CD8, 4-1BB, CD3ζ	Preclinical	[70]
GD2	Ewing sarcoma	PB NK	Retroviral	CD28, 4-1BB, CD3ζ	Preclinical	[71]
Mesothelin	Ovarian carcinoma	NK-92	Lentiviral	CD8, CD28, 4-1BB, CD3ζ	Preclinical	[72]
CD33	Acute myeloid leukemia	NK-92	Lentiviral	CD28, 4-1BB, CD3ζ	Phase I and II trial	[73]
CD123	Acute myeloid leukemia	NK-92	Retroviral	CD28, 4-1BB, CD3ζ	Preclinical	[74]
CS1	Multiple myeloma	NK-92	Lentiviral	CD28, CD3ζ	Preclinical	[75]
CD7	Acute T lymphoblastic leukemia	NK-92MI	Electroporation	CD28, 4-1BB, CD3ζ	Preclinical	[76]
PSMA	Prostate carcinoma	NK-92	Lentiviral	CD28, CD3ζ	Preclinical	[77]
ROBO1	Pancreatic ductal adenocarcinoma	NK-92	Lentiviral	CD8, 4-1BB, CD3ζ	Phase I and II trial	[78]
BCMA	Multiple myeloma	PB NK	Electroporation	CD8, DAP12|CD3ζ	Preclinical	[79]
CD20	Burkitt lymphoma	PB NK	Electroporation	4-1BB, CD3ζ	Preclinical	[80]
PSCA	Prostate adenocarcinoma	YT NK	Lentiviral	CD8, CD28, CD3ζ	Preclinical	[81]
GPA7	Melanoma	NK-92MI	Electroporation	HLA-A2, CD3ζ	Preclinical	[48]
NKG2DL	Multiple myeloma	PB NK	Lentiviral	4-1BB, CD3ζ	Preclinical	[82]
GPC3	Hepatocellular carcinoma	PB NK	Lentiviral	CD8, 4-1BB, CD3ζ	Preclinical	[83]
FR α	Ovarian adenocarcinoma	NK-92	Lentiviral	CD8, CD28, 4-1BB, CD3ζ	Preclinical	[84]
CD276	Neuroblastoma	NK-92	Lentiviral	CD8, CD28, CD3ζ	Preclinical	[85]
CD135	Acute B lymphoblastic leukemia	NK-92	Lentiviral	CD28, CD3ζ	Preclinical	[86]
CD5	Acute T lymphoblastic leukemia	NK-92	Lentiviral	CD8, 2B4, CD3ζ	Preclinical	[60]
PDL1	Head and neck squamous cell carcinoma	haNK	Electroporation	CD8, CD28, FcεR1γ	Preclinical	[87]
TF	Triple-negative breast cancer	NK92MI	Lentivirus	CD28, 4-1BB, CD3ζ	Preclinical	[88]
CD38	Multiple myeloma	KHYG-1	Retroviral	CD8, CD28, 4-1BB, CD3ζ	Preclinical	[89]
CEA	Colorectal carcinoma	NK92MI	Retroviral	CD8, CD3ζ	Preclinical	[90]
CD147	Hepatocellular carcinoma	NK92MI	Retroviral	CD28, 4-1BB, CD3ζ	Preclinical	[91]
WT-1	Leukemia	NK92MI	Retroviral	4-1BB, CD3ζ	Preclinical	[92]
CD4	Acute myeloid leukemia	NK-92	Lentivirus	CD28, 4-1BB, CD3ζ	Preclinical	[93]
c-MET	Hepatocellular carcinoma	PB NK	Lentiviral	CD8, 4-1BB, DAP12	Preclinical	[94]
CD138	Hematologic malignancies	NK-92	Lentiviral	CD8, CD28, 4-1BB, CD3ζ	Preclinical	[95]
CD3	T-Cell lymphoma	NK-92	Lentiviral	CD28, 4-1BB, CD3ζ	Preclinical	[96]

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
