# Peer review of "NK Cells Armed with Chimeric Antigen Receptors (CAR): Roadblocks to Successful Development"

_cells, 2021, doi:10.3390/cells10123390_

Round 1

Reviewer 1 Report

In the current review, Dezfouli et al. aimed to summarize the roadblocks facing gene engineered-NK cells to express chimeric antigen receptor (CAR) with a specific focus on targets selection, CAR design, and gene delivery. The authors used parallelism between CART and CAR-NK cells since later and its clinical application is still in early days, therefore creating speculations in different topics covered by this review. Most subjects are insufficiently covered and skewed into NK cells over more advanced T cell therapies. This review required substantial improvement to add to the body of knowledge already present in the literature. 

Major concerns:

1- The authors unnecessary elaborated on several topics related to CAR designs (scFV, hinge, etc..), antigens (CD19, HER2, etc..) as well as gene transfer methodologies. Most of the text is spent on general knowledge in the field of cell therapy and has been well covered in quality reviews. 

2- Overall, references are not carefully chosen by the authors. They cited several reviews with no mentioning of the original work. Some claims by the authors are not backed with cited references. 

3- What is the rationale for focusing on some targets like CD19, Her2, etc. Many other targets have been described and well-reviewed in the literature. 

4- When describing tumor escape from current CAR T cells, the authors did not accurately describe the current knowledge in the field. For instance, a substantial fraction of relapses after CAR T cell therapy are due to lack of T cells persistence, accidental insertion of CAR gene into tumor cells, antigen-negative escape, antigen low escape, etc... 

5- What do the authors mean by “CAR T cell therapy such as their relatively long lifetime, MHC restriction”? 

6- The authors mention “In the future, advanced scFv structures could be designed to address antigen heterogeneity and prevent tumor escape”. What are these advances scFv structure? If the authors mean bi-specific CAR or dual targeting, these designs are already in the clinic with published reports. 

7- The authors warn that “the popularity of retrovirus in favor of lentivirus for CAR delivery”. (1) what types of retroviruses the authors are warning against? (2) Clinical data with CART have shown otherwise where accidental insertions of the transgene (e.g. insertion in Tet2 locus, CARB with CD19 and CD22) are observed with lentivirus. 

8- In the section on gene delivery, the authors dedicated a section for “CRISPR-Cas9”, CRISPER -CAS 9 system is not a method of gene delivery by itself. This section requires careful description. 

9-Many roadblocks in the CAR NK domain are overlooked ( e.g. "off-the-shelf’ manufacturing, multi-antigen targeting, level of antigen expression, tumor microenvironment, and solid tumor infiltration). 

Author Response

The authors would like to thank the editor and the reviewer for their precious time and valuable comments that will significantly improve the manuscript. We have carefully considered the comments and tried our best to provide point-by-point responses, a complete re-revision of our manuscript and the necessary changes according to the referee's indications. Moreover, the manuscript has been re-checked for English content by one of the authors (AGP), who is a native English speaker.

  • The authors unnecessary elaborated on several topics related to CAR designs (scFV, hinge, etc..), antigens (CD19, HER2, etc..) as well as gene transfer methodologies. Most of the text is spent on general knowledge in the field of cell therapy and has been well covered in quality reviews. 

With respect to the reviewer's comment, the following explanations may clarify the intention of the authors' for this review:

Most recently, CAR cell therapy has shown promising results. Different strategies have been developed to optimize CAR cell therapy to achieve optimal anti-tumor effects with minor side effects. The manuscript focuses on three major steps to optimize CAR cell therapies: choice of tumor-specific targets, optimal design, and efficient CAR cell delivery. The selection of a tumor-specific antigen and the production of an optimized CAR structure are required to improve targeting efficiency and a safer application of CAR cells. The performance of CAR-engineered cells not only depends on the design of the CAR, but also on the delivery system. CARs efficiently target tumor cells when they are highly expressed on the effector cell, which is enabled by a suitable delivery system.

The first CAR cell therapies were developed for T cells, following which other cellular platforms, including NK cells, were used for CAR cell engineering. NK cells have attracted much attention due to their beneficial properties, including a better safety profile and a broader (from the shelf) applicability. A new section #2 (page 3 and 4, lines 129-201) summarizing the advantages and disadvantages of NK cell-based CARs has been included in the revised version of the Ms.

With respect to the specific comment, and also with due respect to the reviewer, we feel that it is important to provide key information relating to the development of CAR T cells and align this with current studies relating to the development of CAR NK cells. It is not possible to achieve this by pointing the reader to other publications, many of which might not be available in full text form.

  • Overall, references are not carefully chosen by the authors. They cited several reviews with no mentioning of the original work. Some claims by the authors are not backed with cited references. 

This comment is well taken. Mostly original articles are now cited in the revised manuscript. Furthermore, additional references have been included as recommended, for example Ref. 14 and 15 (page 2, line 88); Ref.65 (page 7, line 321); Ref. 20 (page 7, line 339).  

  • What is the rationale for focusing on some targets like CD19, Her2, etc. Many other targets have been described and well-reviewed in the literature. 

Table 1 provides an overview of tumor-associated antigens which are presently used for NK cell-based CAR therapies. Targets like CD19 and HER2 were discussed in more detail because they are most commonly used in vitro and in clinical practice and for optimized therapies (i.e., bi-specific targeting) to improve outcome. We therefore believe that they are more relevant to the key elements and aims of this review.

  • When describing tumor escape from current CAR T cells, the authors did not accurately describe the current knowledge in the field. For instance, a substantial fraction of relapses after CAR T cell therapy are due to lack of T cells persistence, accidental insertion of CAR gene into tumor cells, antigen-negative escape, antigen low escape, etc... .

We thank the reviewer for highlighting that tumor escape and relapse are induced by several mechanisms, including lack of T cell persistence, accidental insertion of the CAR gene into tumor cells, antigen-escape, etc. The revised manuscript now makes reference to the fact that  tumor relapse mechanisms after CAR T cell therapy can involve decreased or lost antigen expression (page 4, lines 172-176; page 10, lines 370-374) and low persistence (page 4, lines 185-191). We believe that a more in-depth discussion is beyond the scope of this review which focusses on the development of optimized CAR NK cells.

  • What do the authors mean by "CAR T cell therapy such as their relatively long lifetime, MHC restriction"?

The authors appreciate the clarity issues with this sentence. The sentence has been revised and the biological differences better explained in a new section #2 (page 3 and 4, lines 148-156; page 4, lines 185-191) of the revised MS:

“A prolonged survival of CAR T cells resulting in a long persistence in the body can exert dual effects. Although long‐term persistence of CD19 CAR T cells is important to achieve a durable anti-tumor immunity to prevent tumor recurrence, these cells can cause severe B cell aplasia, increased infectious complications, and cell fratricide due to the shared antigen expression on malignant and non-malignant T cells. Therefore, an enhanced CAR T cell expansion and persistence may act like a double-edged sword [1-3]”.

  • The authors mention "In the future, advanced scFv structures could be designed to address antigen heterogeneity and prevent tumor escape". What are these advances scFv structure? If the authors mean bi-specific CAR or dual targeting, these designs are already in the clinic with published reports. 

We thank the reviewer for this comment. For more clarity, the sentence has been revised accordingly (page 5, lines 226-228. The "advanced scFv structures" are attributed to the bi-specific targeting capacity of CARs. This approach is presently applied pre-clinically and in early clinical trials. The rareness of tumor-specific antigens, especially in solid malignancies, impairs the effectiveness of CAR cell therapies, which also requires the design of more advanced scFvs to compensate for some inherent shortcomings of CAR therapies. Therefore, further studies are necessary to identify the best targeting moieties for tumor antigens.

  • The authors warn that "the popularity of retrovirus in favor of lentivirus for CAR delivery". (1) what types of retroviruses the authors are warning against? (2) Clinical data with CART have shown otherwise where accidental insertions of the transgene (e.g. insertion in Tet2 locus, CARB with CD19 and CD22) are observed with lentivirus. 

The authors appreciate the clarity issues with corresponding revision.

The increasing popularity of lentivirus-based transduction are due to published advantages such as lower rate of genotoxicity potential, and transducing ability for both dividing and non-dividing cells [4-6]. To date, 14 primary NK cells and 44 NK cell lines have been reported to have been successfully transduced with lentiviral vectors, whereas 16 primary NK cells and 19 NK cell lines have been reported to have been successfully transduced with retroviral vectors [7].

The semi-random integration of lentiviral vectors still carries the risk of insertional mutagenesis and dysregulation, but to a lesser extent than retroviral vectors and may provide a safer alternative (page 15, lines 621-623). Montini et al. showed that the difference can be attributed to the degree and integration site selection between lentiviral and retroviral vectors [4] (this study has been added to the revised version of the Ms on page15, lines  623-624 with Ref. #165).

The correlation between the lentiviral vector-integration site and the therapeutic outcome of CAR T cell has been also added to the Ms with referring to the Tet2 integration in genomic DNA by lentivirally transduced CAR T cells [8] (page 15, lines 624-626 with Ref. #169).

  • In the section on gene delivery, the authors dedicated a section for "CRISPR-Cas9", CRISPER -CAS 9 system is not a method of gene delivery by itself. This section requires careful description.

As the referee correctly mentioned, CRISPER-CAS9 is a gene-editing tool in the field of CAR cell engineering. The authors clarified this aspect in the revised version of the Ms (page 19, line 832) by emphasizing its editing potential to increase the efficacy of the CAR cells. Furthermore, the potential of the technique has been outlined in the Gene Editing part of Figure 2.

  • Many roadblocks in the CAR NK domain are overlooked (e.g. "off-the-shelf' manufacturing, multi-antigen targeting, level of antigen expression, tumor microenvironment, and solid tumor infiltration). 

Although CAR cell therapy is a novel and promising strategy to fight cancer, some roadblocks limit the developmental path to achieve the optimum peak of efficiency. The limiting factors can emerge during all steps from manufacturing to clinical application. Amongst others, in this review, we concentrated on topics related to antigen recognition, design, and delivery of CARs. The final treatment outcome can be affected by each step, including selecting a specific tumor antigen, the design of the CAR (with a precise targeting and sufficient stimulatory effect), and the system for CAR transfer into NK cells. Moreover, several other challenges (as mentioned by the referee) e.g., tumor resistance (due to the decrease or loss of tumor antigen), low infiltration rate to the solid tumors (due to the limiting factors in the tumor microenvironment)[9,10], and finding an “off-the-shelf” source for CAR cell manufacturing [11] can also impair the therapy response. These factors have been discussed in more detail in other reviews which are also cited in the revised version of the Ms with ref. #16-18.

  1. Ghorashian, S.; Kramer, A.M.; Onuoha, S.; Wright, G.; Bartram, J.; Richardson, R.; Albon, S.J.; Casanovas-Company, J.; Castro, F.; Popova, B. Enhanced CAR T cell expansion and prolonged persistence in pediatric patients with ALL treated with a low-affinity CD19 CAR. Nature medicine 2019, 25, 1408-1414.
  2. Mamonkin, M.; Mukherjee, M.; Srinivasan, M.; Sharma, S.; Gomes-Silva, D.; Mo, F.; Krenciute, G.; Orange, J.S.; Brenner, M.K. Reversible transgene expression reduces fratricide and permits 4-1BB costimulation of CAR T cells directed to T-cell malignancies. Cancer immunology research 2018, 6, 47-58.
  3. Hill, J.A.; Li, D.; Hay, K.A.; Green, M.L.; Cherian, S.; Chen, X.; Riddell, S.R.; Maloney, D.G.; Boeckh, M.; Turtle, C.J. Infectious complications of CD19-targeted chimeric antigen receptor–modified T-cell immunotherapy. Blood, The Journal of the American Society of Hematology 2018, 131, 121-130.
  4. Montini, E.; Cesana, D.; Schmidt, M.; Sanvito, F.; Ponzoni, M.; Bartholomae, C.; Sergi, L.S.; Benedicenti, F.; Ambrosi, A.; Di Serio, C. Hematopoietic stem cell gene transfer in a tumor-prone mouse model uncovers low genotoxicity of lentiviral vector integration. Nature biotechnology 2006, 24, 687-696.
  5. Micucci, F.; Zingoni, A.; Piccoli, M.; Frati, L.; Santoni, A.; Galandrini, R. High-efficient lentiviral vector-mediated gene transfer into primary human NK cells. Experimental hematology 2006, 34, 1344-1352.
  6. Alici, E.; Sutlu, T.; Dilber, M.S. Retroviral gene transfer into primary human natural killer cells. In Genetic Modification of Hematopoietic Stem Cells; Springer: 2009; pp. 127-137.
  7. Gong, Y.; Wolterink, R.G.K.; Wang, J.; Bos, G.M.; Germeraad, W.T. Chimeric antigen receptor natural killer (CAR-NK) cell design and engineering for cancer therapy. Journal of Hematology & Oncology 2021, 14, 1-35.
  8. Fraietta, J.A.; Nobles, C.L.; Sammons, M.A.; Lundh, S.; Carty, S.A.; Reich, T.J.; Cogdill, A.P.; Morrissette, J.J.; DeNizio, J.E.; Reddy, S. Disruption of TET2 promotes the therapeutic efficacy of CD19-targeted T cells. Nature 2018, 558, 307-312.
  9. Rafei, H.; Daher, M.; Rezvani, K. Chimeric antigen receptor (CAR) natural killer (NK)‐cell therapy: leveraging the power of innate immunity. British journal of haematology 2021, 193, 216-230.
  10. Lamb, M.G.; Rangarajan, H.G.; Tullius, B.P.; Lee, D.A. Natural killer cell therapy for hematologic malignancies: successes, challenges, and the future. Stem Cell Research & Therapy 2021, 12, 1-19.
  11. Morgan, M.A.; Büning, H.; Sauer, M.; Schambach, A. Use of cell and genome modification technologies to generate improved “off-the-shelf” CAR T and CAR NK cells. Frontiers in immunology 2020, 11, 1965.

Reviewer 2 Report

The authors summarize the principle of NK cell CAR expressing cells and list preclinical and clinical trials utilising NK CARs. Further, TAAs are discussed in the context of NK CARs and different gene deliveries are outlined as a procedural basis of NK CAR manufacturing.

The review is well-written and structured.

The review summarizes important information around NK CARs in general, however the biological background and comparison to T cell CARs is underrepresented. Thus, the review lacks the explanation why and how NK cells could become a new potent therapeutic in the fight of cancer.

Please find my recommendations that shall be addressed to substantially improve this overall valuable review article and helps to connect the information in a general context of CAR therapy rather than just listing aspects of NK CAR approaches.

By adding some thoughts about where to place NK CAR therapy at the moment and what work has to be done to make it an alternative to T CARs will substantially improve the manuscript and will address the interests of a broader readership.

Minor:

  1. no acronyms in abstract
  2. missing of in line 57
  3. relatively long lifetime in line 72 (not a good expression)
  4. relatively short lifespan in line 91/92 (not a good expression)
  5. must be modified in line 182
  6. Table 1 gives a good overview on NK CAR studies - critical concluding remarks should be made on the different CAR products with regard to CAR generation, commonalities and differences
  7. in line 269-272 (see above) should be commented on - why?
  8. Too many citations for the length of the manuscript 

Major

  1. ICANS and CRS in line 73 (What is the mechanism of ICANS and CRS? How is the managment and what are the clinical problems? What is the dependency on the costimulatory signaling? Why are NK cells more suitable? I would suggest it's the lack of proliferative potential and subsequent toxicity mediated by cytokines. The direct mechanism of neurotoxic effect also depends on the CAR affinity, e.g. compare high-affinity FMC63 CAR versus humanized low affinity CAR (nature medicine)
  2. if you bring up GvHD and autoimmune response the mechanisms should be commented on in line 94. Do CAR T cells cause GvHD? I don't think there is sufficient proof. However, they may trigger aGvHD as well as support cGvHD which we have observed in the past.
  3. There should be a significant section on the biological assets of NK cells versus T cells. This has major implications on how and why NK cells may be an interesting cell subset for CAR therapy. The main issue and question should be addressed around the engraftment and proliferation capacity as this dictates the overall anti-cancer activity.
  4. The comparison of NK- and T-cell signaling is interesting for the reader and should be addressed more in depth. An illustration on the downstream effects should be made accessible to the reader
  5. Figure No 1 is of very limited value and could be substituted by a figure demonstrating the features of NK cell versus T cell CAR expressnig effector cells with regard to naïve and modified cells. The impact of the CAR generation protocol should be considered as well.
  6. The discussion of various TAA is interesting but again more interesting is the general comparison of NK versus T cells with regard to behaviour, performance, safety etc. and if possible comment on why
  7. The roadblocks in making NK cells a potent therapeutic lies most in the biology of NK cells rather than in technical obstacles. It's interesting to outline the different gene delivery technologies but again the main question remains unanswered. What has to happen to make NK cells an alternative to T cells? NK cells are by far the more potent subset in a direct comparison but at this stage T cells have the ability to exponentially outgrow cancer cells which NK cells do not. If NK cells had the same properties, e.g. engraftment, exponential antigen-specific proliferation, persistence they would be an alternative to T cells. 

As a recommendation to improve the manuscript, Malcolm Brenner and colleagues have extensively studied NK cells and NK CARs as well as artificial cell subsets which demonstrated both safety and efficacy. Maybe a short reflection on NK, NKT and T cells would highlight the limitations and great potential of NK cells or artificial NK-like cell subsets more appropriately. 

Author Response

The authors would like to thank the editor and the reviewer for their precious time and valuable comments that will significantly improve the manuscript. We have carefully considered the comments and tried our best to provide point-by-point responses, a complete re-revision of our manuscript and the necessary changes according to the referee's indications.

The authors summarize the principle of NK cell CAR-expressing cells and list preclinical and clinical trials utilising NK CARs. Further, TAAs are discussed in the context of NK CARs and different gene deliveries are outlined as a procedural basis of NK CAR manufacturing.

The review is well-written and structured.

The review summarizes important information around NK CARs in general, however the biological background and comparison to T cell CARs is underrepresented. Thus, the review lacks the explanation why and how NK cells could become a new potent therapeutic in the fight of cancer.

We thank the reviewer for this comment and, as suggested, we have included a new section #2 (pages 3 and 4, lines 129-201) titled "NK Cells- a promising cellular platform for CAR engineering" in the revised Ms. This discusses the potential properties of the NK cells, which make it a suitable candidate for CAR cell-based immunotherapy.

Please find my recommendations that shall be addressed to substantially improve this overall valuable review article and helps to connect the information in a general context of CAR therapy rather than just listing aspects of NK CAR approaches.

By adding some thoughts about where to place NK CAR therapy at the moment and what work has to be done to make it an alternative to T CARs will substantially improve the manuscript and will address the interests of a broader readership.

Minor:

  1. no acronyms in abstract.

It has been revised.

  1. missing of in line 5.

It has been corrected.

  1. relatively long lifetime in line 72 (not a good expression).

It has been re-written in more detail in section #2 (pages 3 and 4, lines 129-201).

  1. relatively short lifespan in line 91/92 (not a good expression).

It has been re-written in more detail in section #2 (pages 3 and 4, lines 129-201).

  1. must be modified in line 182.

It has been changed.

  1. Table 1 gives a good overview on NK CAR studies - critical concluding remarks should be made on the different CAR products with regard to CAR generation, commonalities and differences,

The aim of table 1 is to provide the general information from published manuscripts about tumor target, NK cell source, CAR structure, delivery, and the stage of the experiments (fitting more to our aims).

In this case, a conclusion statement has been added to the manuscript in page 7, lines 328-334.

  1. in line 269-272 (see above) should be commented on - why?

According to the referee's great comment, section #2 (pages 3 and 4, lines 129-201) has been added to the revised Ms answering why and how NK cells could become a new potent therapeutic in the fight against cancer.

  1. Too many citations for the length of the manuscript 

The authors have reconsidered the referencing. Furthermore, mostly original articles are now cited in the revised manuscript. However, due to adding a new section #2 (pages 3 and 4, lines 129-201) and comments of reviewer 1 additional references have been included.

Major

  1. ICANS and CRS in line 73 (What is the mechanism of ICANS and CRS? How is the management and what are the clinical problems? What is the dependency on the costimulatory signaling? Why are NK cells more suitable? I would suggest it's the lack of proliferative potential and subsequent toxicity mediated by cytokines. The direct mechanism of neurotoxic effect also depends on the CAR affinity, e.g. compare high-affinity FMC63 CAR versus humanized low affinity CAR (nature medicine)

We thank the reviewer for this valuable comment - we have added section #2 (pages 3 and 4, lines 129-201) to the revised Ms addressing all the queries.

Furthermore, the original manuscript suggested by the referee provides valuable data, reference to which has been added in the corresponding paragraphs (page 4, lines 183-185) and (page 5, lines 222-226) (ref. # 31).

  1. If you bring up GvHD and autoimmune response the mechanisms should be commented on in line 94. Do CAR T cells cause GvHD? I don't think there is sufficient proof. However, they may trigger aGvHD as well as support cGvHD which we have observed in the past.

More details regarding GVHD have been included in the newly added section #2 (page 4, lines 191-196) in order to clarify the important point mentioned by the referee:

"The allogeneic “off-the-shelf” CAR T cell products frequently induce GVHD, a clinically immuno-incompatibility syndrome which can lead to substantial morbidity and mortality, due to HLA mismatches between donor and recipient [1]. However, allogeneic NK cells exhibit a better safety profile which permits their use from healthy universal donors. The shorter lifespan of NK cells may reduce autoimmunity and long-term side effects [2,3]."

  1. There should be a significant section on the biological assets of NK cells versus T cells. This has major implications on how and why NK cells may be an interesting cell subset for CAR therapy. The main issue and question should be addressed around the engraftment and proliferation capacity as this dictates the overall anti-cancer activity.

According to the referee’s comments, we have included section #2 (pages 3 and 4, lines 129-201) which covers the biological capacities of NK cells that support their use as promising CAR-based therapeutics.

  1. The comparison of NK- and T-cell signaling is interesting for the reader and should be addressed more in depth. An illustration on the downstream effects should be made accessible to the reader.

Antigen-CAR interactions trigger an intracellular signaling cascade and subsequent immune cell activation against the antigen-expressing target cells. As the referee correctly mentions, the interesting aspects of downstream signaling have not been covered in this review due to space issues. We have briefly covered this on page 6 in lines 275-280 and supported this statement with Ref. #33 (published by Prof. Brenner). Downstream signaling of CAR-NK cells (in combination with the CRISPR Cas9 technique) is a topic of our current research project and a review manuscript that we are currently working on intensively.

  1. Figure No 1 is of very limited value and could be substituted by a figure demonstrating the features of NK cell versus T cell CAR expressing effector cells with regards to naïve and modified cells. The impact of the CAR generation protocol should be considered as well.

Figure 1 shows the general activity of CAR NK cells followed by infusion to the body and a general illustration of CAR structure composed of the different main domains.

We would like to respectfully mention that the aim of this study was not to compare CAR NK cells with CAR T cells. The authors are aware that while several CAR T cells are already approved by the FDA, CAR NK cell-based immunotherapies are still in their infancy. However, due to some advantages of NK cells (as added in section #2, pages 3 and 4, lines 129-201), NK cells derived from different sources might provide promising platforms for CAR engineering. The main goal of this manuscript was therefore to describe different strategies for optimizing these immune cell candidates for an efficient and safe CAR-based therapy.

  1. The discussion of various TAA is interesting but again more interesting is the general comparison of NK versus T cells with regard to behaviour, performance, safety etc. and if possible comment on why.

According to the referee's comment, a general comparison of NK cells versus T cells has been included in the new section #2 (pages 3 and 4, lines 129-201) of the revised version.

  1. The roadblocks in making NK cells a potent therapeutic lies most in the biology of NK cells rather than in technical obstacles. It's interesting to outline the different gene delivery technologies but again the main question remains unanswered. What has to happen to make NK cells an alternative to T cells? NK cells are by far the more potent subset in a direct comparison but at this stage T cells have the ability to exponentially outgrow cancer cells which NK cells do not. If NK cells had the same properties, e.g. engraftment, exponential antigen-specific proliferation, persistence they would be an alternative to T cells. 

We appreciate the referee's great recommendations. Accordingly, section #2 (pages 3 and 4, lines 129-201) covering the biologic properties and unique advantages of CAR-NK cells over CAR-T cells, has been added to the manuscript.

As a recommendation to improve the manuscript, Malcolm Brenner and colleagues have extensively studied NK cells and NK CARs as well as artificial cell subsets which demonstrated both safety and efficacy. Maybe a short reflection on NK, NKT and T cells would highlight the limitations and great potential of NK cells or artificial NK-like cell subsets more appropriately. 

We appreciate the great recommendation of the referee. Prof. Brenner and his lab have indeed published many very interesting manuscripts regarding CAR-engineered cells. Some of his publications [4-7] have been added into the revised version of the Ms.

  1. Depil, S.; Duchateau, P.; Grupp, S.; Mufti, G.; Poirot, L.' Off-the-shelf’allogeneic CAR T cells: development and challenges. Nature reviews Drug discovery 2020, 19, 185-199.
  2. Liu, E.; Marin, D.; Banerjee, P.; Macapinlac, H.A.; Thompson, P.; Basar, R.; Nassif Kerbauy, L.; Overman, B.; Thall, P.; Kaplan, M. Use of CAR-transduced natural killer cells in CD19-positive lymphoid tumors. New England Journal of Medicine 2020, 382, 545-553.
  3. Morgan, M.A.; Büning, H.; Sauer, M.; Schambach, A. Use of cell and genome modification technologies to generate improved “off-the-shelf” CAR T and CAR NK cells. Frontiers in immunology 2020, 11, 1965.
  4. Mamonkin, M.; Mukherjee, M.; Srinivasan, M.; Sharma, S.; Gomes-Silva, D.; Mo, F.; Krenciute, G.; Orange, J.S.; Brenner, M.K. Reversible transgene expression reduces fratricide and permits 4-1BB costimulation of CAR T cells directed to T-cell malignancies. Cancer immunology research 2018, 6, 47-58.
  5. Watanabe, N.; Bajgain, P.; Sukumaran, S.; Ansari, S.; Heslop, H.E.; Rooney, C.M.; Brenner, M.K.; Leen, A.M.; Vera, J.F. Fine-tuning the CAR spacer improves T-cell potency. Oncoimmunology 2016, 5, e1253656.
  6. Savoldo, B.; Rooney, C.M.; Di Stasi, A.; Abken, H.; Hombach, A.; Foster, A.E.; Zhang, L.; Heslop, H.E.; Brenner, M.K.; Dotti, G. Epstein Barr virus–specific cytotoxic T lymphocytes expressing the anti-CD30ζ artificial chimeric T-cell receptor for immunotherapy of Hodgkin disease. Blood, The Journal of the American Society of Hematology 2007, 110, 2620-2630.
  7. Shaw, A.R.; Porter, C.E.; Watanabe, N.; Tanoue, K.; Sikora, A.; Gottschalk, S.; Brenner, M.K.; Suzuki, M. Adenovirotherapy delivering cytokine and checkpoint inhibitor augments CAR T cells against metastatic head and neck cancer. Molecular Therapy 2017, 25, 2440-2451.

Round 2

Reviewer 1 Report

No further comments 

Author Response

The authors would like to thank the reviewer for his/her precious time and valuable comments.

Reviewer 2 Report

The authors addressed most of the comments.

I do not particularly like and appreciate Figure 1 as written in my previous comments as it does not add any value to the manuscript.

I disagree about the GvHD section on CAR T cells as there is no proof to date that CAR T cells are the causing cell fraction of GvHD. In my own clinical experience allogeneic CAR T cells do not cause GvHD even in the haploidentical setting. 

What we have observed instead was GvHD triggering of GvHD in the context of CAR T cell therapy in which however CAR T cells were not the infiltrating cells in GvHD damaged tissue.

Further native NK CAR expressing cells are presented as an alternative effector cell subset without a clear proposal how these cells shall manage the most critical aspects of CAR function which is exponential proliferation. 
